# Alkaline Degradation of Plant Fiber Reinforcements in Geopolymer: A Review

**DOI:** 10.3390/molecules28041868

**Published:** 2023-02-16

**Authors:** Chun Lv, Jie Liu

**Affiliations:** 1College of Architecture and Civil Engineering, Qiqihar University, Qiqihar 161006, China; 2College of Light Industry and Textile, Qiqihar University, Qiqihar 161006, China; 3Engineering Research Center for Hemp and Product in Cold Region of Ministry of Education, Qiqihar 161006, China

**Keywords:** plant fiber, geopolymers, alkaline degradation, compatibility, absorbency, ductility

## Abstract

Plant fibers (PFs), such as hemp, Coir, and straw, are abundant in resources, low in price, light weight, biodegradable, have good adhesion to the matrix, and have a broad prospect as reinforcements. However, the degradation of PFs in the alkaline matrix is one of the main factors that affects the durability of these composites. PFs have good compatibility with cement and the geopolymer matrix. They can induce gel growth of cement-based materials and have a good toughening effect. The water absorption of the hollow structure of the PF can accelerate the degradation of the fiber on the one hand and serve as the inner curing fiber for the continuous hydration of the base material on the other. PF is easily deteriorated in the alkaline matrix, which has a negative effect on composites. The classification and properties of PFs, the bonding mechanism of the interface between PF reinforcements and the matrix, the water absorption of PF, and its compatibility with the matrix were summarized. The degradation of PFs in the alkaline matrix and solution, drying and wetting cycle conditions, and high-temperature conditions were reviewed. Finally, some paths to improve the alkaline degradation of PF reinforcement in the alkaline matrix were proposed.

## 1. Introduction

In recent decades, the fiber reinforcements of cement and geopolymer composites have mainly used traditional fibers [1]. Traditional fibers include metal fibers [2,3], inorganic fibers [4,5], and synthetic fibers [6,7]. The production of traditional metal fibers and inorganic fibers requires a large amount of resources and high cost. Synthetic fibers are easily polluting the environment in the production process, which is inconsistent with the requirements of sustainable development [8,9]. However, plant fiber (PF) is the most widely found fiber in nature. It has many advantages, such as its wide source, low price, simple manufacturing process, energy savings, and environmental protection, and it has gradually attracted the attention of researchers [10,11].

PFs not only have abundant resources and low cost, but they are also light weight, have good mechanical properties, and strong adhesion to the matrix. At the same time, compared to other types of fibers, PFs are biodegradable, reducing dependence on non-renewable resources and reducing greenhouse gas and pollutant emissions. Cement and geopolymers are characterized by weak tensile strength and low toughness. These problems can be alleviated by using PFs as reinforcements of composites [12]. Therefore, in recent years, PFs as reinforcements of composites have been studied by scientists and gradually applied in engineering projects [13].

PF is mainly composed of cellulose, hemicellulose, lignin, and other substances, its main component being cellulose. The geometric characteristics, mechanical properties, mixing rate, and bonding properties between PFs and the matrix have important effects on the strength and toughness of composites. In commonly used conventional fibers, steel fiber lacks an induction effect on the growth of the hydration gel in the transition zone of the matrix interface [14]. Carbon fiber is prone to agglomeration in the matrix, and its price is relatively high [15]. Synthetic polymer fiber has poor compatibility with the matrix and poor dispersion in the matrix [16,17]. In comparison to the above traditional fibers, the specific surface area and length diameter of PFs are larger, and they have good toughness. PF reinforcements have a good bonding ability with the matrix, are evenly dispersed in the matrix, and have good compatibility with the matrix, and they can fill and bridge the pores and cracks of the matrix [18]. At the same time, the addition of PFs also reduces the density of the composite material, improves its tensile strength, inhibits the expansion of microcracks in the matrix, and improves the impact resistance of the cement-based composite [19,20]. However, PFs have relatively low resistance to degradation in alkaline environments, and their long-term strength will decrease when used as a reinforcement for substrates exposed to harsh environmental conditions. In the alkaline mineral environment, the degradation of PF leads to the deterioration of the durability of composite materials, which weakens the strengthening effect of the fiber and has an important influence on the long-term service life of the structure [21,22]. The degradation of PF in the matrix is the main problem, which must be solved in its engineering application [23].

There are many research cases on the deterioration of PF reinforcement in matrix materials. Deterioration of PF-reinforced composites is closely related to the main chemical composition of the fiber itself. In addition to cellulose, hemicellulose, and lignin components, other components of PF include wax, pectin, and various water-soluble components. Cellulose is a network of fine cellulose and lignin embedded in amorphous crystal microfibers, which provide structural support for the fibers. Filho et al. [24] described the dissolution of lignin and hemicellulose in cement pore solution and the alkaline hydrolysis of cellulose molecules, leading to degradation of molecular chains and reduction in the degree of polymerization and tensile strength, which are the main aging mechanisms of natural fibers in cement-based composites. In addition, lime crystallization in the fiber cavity and midlamella also reduces the flexibility and strength of the fiber [25]. Mohr et al. [26] proposed a three-part progressive degradation mechanism during the drying and wetting cycles. Fiber embrittlement caused by mineralization of the fiber cell wall was manifested as insufficient resilience. Filho et al. [27] studied the adsorption of calcium and hydroxyl ions, fiber mineralization, and degradation of cellulose, hemicellulose, and lignin in PF in cement composites due to the presence of calcium hydroxide. However, the inherent degradation kinetics and mechanism of PF and its three main components, cellulose, hemicellulose, and lignin, in the cement solid phase and pore solution have not been clarified, and further research is needed. In general, excess plant fibers adversely affect the compressive strength of the matrix. However, an appropriate amount of plant fibers can be evenly distributed in the matrix to enhance the compactness of the matrix, thus reducing porosity and cracks and improving compressive strength [28,29].

The deterioration of the properties of PF-reinforced cement and geopolymer composites is due to weathering, alkaline erosion, and other external reasons. This external reason is closely related to its internal factors, such as the influence of compatibility between PFs and the matrix interface region and the change in fiber volume [30]. These factors cause the degradation of PFs due to physical, biological, mechanical, and chemical attacks. Especially when exposed to an outdoor environment, PF-reinforced composites show a higher risk of deterioration under the influence of environmental factors, such as ultraviolet radiation, temperature, humidity, mechanical stress, chemical agents, and microbial activity [31,32,33]. From the existing studies, there are few reviews on the degradation of PFs in the geopolymer matrix. In this paper, on the basis of the properties of PFs, combined with the water absorption of PFs and their compatibility with the matrix, the bonding mechanisms of PF-reinforced geopolymer composites were studied. The matrix toughness and PFs degradation in alkaline solution, drying and wetting cycles, and high-temperature environment were studied. Finally, the effects of nano-material addition and fiber chemical modification on geopolymer composites were analyzed. The objective of this study is to improve the degradation of plant fiber by reducing the alkaline environment of the matrix, improving fiber properties and other perspectives.

## 2. Classification and Mechanical Properties of PFs

PFs exist widely in agricultural residues, mainly composed of cellulose, hemicellulose, lignin, pectin, wax, and some water-soluble materials [34]. Cellulose, hemicellulose, and lignin in PF raw materials mainly exist in plant cells, wherein cellulose is the skeletal substance of the fiber, and lignin and hemicellulose are dispersed in and around the fiber in the form of containing substances. The cellulose fibers commonly used for geopolymeric reinforcement include bast fibers, straw fibers, leaf fibers, stem fibers [35,36,37], and the main representative PFs are shown in Figure 1.

The density, tensile strength, tensile modulus, and elongation at break of different types of PFs are different, and their mechanical properties are shown in Table 1.

**Table 1 molecules-28-01868-t001:** Mechanical properties of commonly used PFs.

Fiber Type	Fiber Name	Density /(g·cm^−3^)	Tensile Strength /MPa	Tensile Modulus/GPa	Elongation/%	Ref.
Bast	Flax	1.50	800–1 500	27.6–80.0	1.2–3.2	[38]
	Hemp	1.48	550–900	70.0	2.0–4.0	[39]
	Jute	1.50	600	10.0–30.0	1.5–1.8	[40,41]
	Kenaf	1.45	930	53.0	1. 6	[42]
	Ramie	1.50	220–938	44.0–128.0	2.0–3.8	[43]
Leaf	Abaca	1.50	400	12.0	3.0–10.0	[44]
	Sisal	0.86	606	15.4	4.1	[45]
	Banana	1.35	600	17.9	3.4	[43]
	Pineapple	1.43	413–1 627	34.5–82.5	1.6	[46]
Fruit	Coir	1.50	500	4.0–6.0	30.0	[40,43]
Wood	Soft wood	1.50	1 000	40.0	4.4	[47]
Grass	Bamboo	1.10	500	35.9	1.4	[47]
Seed	Cotton	1.60	287–597	5.5–12.6	7.0–8.0	[47]

A single PF consists of several basic fibrils bound together by a pectin substance [48]. Cellulose disose is a linear polymer composed of glucose subunits linked by β-1,4 bonds. Fibrils are supported by insoluble chains of microfibers, which are held together by numerous intramolecular and intermolecular hydrogen bonds. Their good tensile resistance enables plant cells to withstand water penetration, which is also the main factor for PFs to be able to withstand external forces [49]. The cell wall structure of PFs endows them with excellent mechanical properties and low density, which also leads to low resistance to lignin and hemicellulose and poor durability in the cement matrix in a high-alkali environment [50].

## 3. PF-Reinforced Geopolymer Composites

In the process of service of PF-reinforced composites, they often face the influence of external environment, especially the influence of water and temperature. After moisture is absorbed, the dimensional stability and mechanical properties of composite materials will decrease. Water molecules not only affect the properties of PFs but also destroy the cohesiveness of the interface between the fiber and the matrix. On the one hand, the volume expansion of PF after moisture absorption produces cracks, which damage the original interface, affecting not only the effective transfer of external forces but also absorbing more water. On the other hand, hydrogen bonds are formed between PFs and water molecules, which is not conducive to the compatibility between the fibers and the matrix [51,52].

### 3.1. Interface Bonding Mechanisms between PFs and Matrix

The PF-reinforced geopolymer composite is composed of fiber and a geopolymer matrix with different properties. The contact surface of the PF and the geopolymer matrix forms an interface. The interface of the composite material is an extremely complex microstructure, including the geometric surface and transition region where the matrix and the fiber interact [53,54]. The best performance of geopolymer composites can be achieved by adjusting the interfacial adhesion state and optimizing the properties of the interfacial layer between the fiber and the matrix. The interface bonding forms of the fiber and the matrix generally include interdiffusion, electrostatic adhesion, chemical bonding, and mechanical interlocking [55], as shown in Figure 2.

According to the microstructure of the fiber and geopolymer bonding, the surface of PF at the microscopic level usually has a certain roughness, and the interface bonding is mainly in the form of mechanical interlocking. Figure 3a shows fiber bridge cracks and fiber debonding, and Figure 3b shows fiber adhesion to the matrix and fiber fracture.

### 3.2. Water Absorption of PF-Reinforced Geopolymers

The hygroscopic process of PF-reinforced geopolymer composites is closely related to fiber reinforcements, the matrix, and the interface. The water absorption process of PFs is complex and mainly based on the diffusion of water molecules. The water absorption of PF-reinforced polymer composites mainly takes three forms. In the first form, water molecules are diffused along the chain of matrix molecules with small gaps; the other form is the capillary action of the interfacial microgap. The third form is the diffusion of water molecules in interfacial cracks caused by fiber expansion. The reason why water molecules can enter the interior of composite materials is that there are micropores in the interior of composite materials through which water molecules diffuse to the interior of composite materials with low concentration. On the other hand, it is related to the molecular polarity, type, and number of functional groups of composite materials [56].

PF molecules contain a lot of hydroxyl, which can bind to polar water molecules. Different types of PFs absorb water differently. More et al. [40] found that the water absorption of coir, jute, and sisal fibers was 130–180%, 20–40%, and 75–80%, respectively. Wei et al. [57] found that sisal fiber water absorption in Ca(OH)_2_ solution increased by 5% compared to ordinary water. In addition, the moisture absorption capacity of the fibers was positively correlated with the ambient temperature.

Due to repeated expansion and contraction of PFs under the action of water, stress is generated inside the fibers, and microcracks are formed [40]. When free water enters PF cells, hydrogen bonds are formed between water and PF, so as to compound the fiber and matrix interface under the action of water [58,59]. In addition to alkaline degradation, the high hygroscopic properties of PFs also create favorable conditions for biodegradation. Chen et al. [60] found that when more water was absorbed by fibrocytes, bound water began to increase, while free water decreased. The soluble component in the fiber dissolves, resulting in the desticking and delamination of the fiber. Methacanon [61] et al. studied the hygrometric properties of the fibers of sisal, reed, water hyacinth, and rose leaf under different relative humidities of air. The hygroscopic property of the fiber increased with increasing relative humidity. Water hyacinth absorbed almost eight times more water at 97% relative humidity than at 75% relative humidity. When relative humidity increased to 97%, the moisture absorption rate of sisal, reed, and rose leaf fibers increased by 4%.

However, as a reinforcement of cement and the geopolymer matrix, the absorbency of the cavity structure of PF, on the one hand, accelerates the degradation of the fiber in the matrix; on the other hand, it can also serve as an internal curing fiber for continuous hydration of the matrix, which can promote the strength growth of the matrix. This is the case because PF is different from other synthetic fibers and traditional fiber advantages.

### 3.3. Compatibility between PF and Geopolymer Matrix

As mentioned above, the formation of hydrogen bonds between PF materials and water molecules is not conducive to the interface compatibility between the fiber and geopolymer. Because of the characteristics of physical structure and chemical composition of natural fiber materials, water is easily absorbable in the surrounding environment, which is not conducive to the processing of composite materials and will lead to poor compatibility with the interface between the polymer matrix. The interfacial properties of different materials have a great influence on the properties of composites. The key technology of interfacial control is to improve the bonding state and the properties of the interfacial layer between fiber reinforcement and the matrix. Na et al. [62] soaked a 1 mol/L CaCl_2_ solution in alkali-treated kenaf fiber, which improved the compatibility between the fiber and the matrix, increasing the bending strength by 69.1% and toughness by 473%. Camargo et al. [37] studied the compatibility of the geopolymer and PFs according to the method of compatibility between PFs and cement-based materials. Yuanita et al. [63] studied the effect of alkalization treatment on the compatibility between PF and the matrix. Hachmi et al. [64] studied the influence of 12 PFs on cement hydration and their compatibility indices. Tan et al. [65] proved that both geopolymers and PFs had good compatibility. Compared to wood fiber, non-wood fiber is less compatible with geopolymers. As can be seen from the polymerization temperature curve of the PF geopolymer, the hydration temperature of the PF-reinforced geopolymer is lower than that of the pure geopolymer, and its maximum temperature is delayed compared to that of the pure geopolymer, indicating that the polymerization of PF is inhibited, as shown in Figure 4. After the temperature peak appeared, the geopolymerization curve of PF-reinforced geopolymers decreased more gently than that of pure geopolymers. The good compatibility between PF and the geopolymer matrix results in a better interfacial bonding performance between the fiber and the matrix, thus delaying the degradation behavior of fiber.

Generally, PFs and geopolymers have good compatibility. However, cement is slightly different from geopolymers. Wei et al. [66] studied the compatibility of 38 kinds of wood with ordinary Portland cement (OPC) using isothermal calorimetry. According to the test results, 24 kinds of wood could regulate the hydration reaction of cement.

## 4. Degradation Behavior of PFs in Geopolymer Matrix

### 4.1. Mechanism of Degradation of PF in the Matrix

The compatibility between the geopolymer matrix and the PF reinforcement makes the interfacial bond of the geopolymer matrix good in the service process and can delay the degradation of the fiber. The cellulose, hemicellulose, and lignin of PF constitute the supporting skeleton of the plant body. Cellulose forms the microfibers that form the reticular skeleton of the cell walls of the fibers, while hemicellulose and lignin act as binders and fillers between fibers and microfibers. There are different types of degradation of PF, including acid hydrolysis, oxidative, alkaline, microbial, thermal, mechanical, and so on. Cellulose degradation is possible under various conditions. The degradation of PF in the cement or the geopolymer matrix is mainly alkaline degradation. Although PFs have good compatibility with geopolymers, they also exhibit degradation behavior due to the influence of alkaline activators in the geopolymers.

The highly complex arrangement of cellulose microfibril makes cellulose fibers more resistant to chemical and biological attacks [67]. Although cellulose degrades into chemicals to some extent, it is highly resistant to hydrolysis, oxidants, and strong bases. Hemicellulose, on the other hand, is highly hydrophilic, readily hydrolyzes in weak acids and bases, and degrades easily when exposed to heat and biological attack [68,69]. The amorphous components of PFs will degrade to different degrees when exposed to alkaline environment, as shown in Figure 5.

Wei et al. [70,71] analyzed the alkaline degradation behavior of PFs in the matrix by studying PF-reinforced cement-based composites. First, the degradation of lignin and part of hemicellulose is followed by the complete degradation of hemicellulose, which damages the integrity of the PF cell wall and then strips the cellulose microfibrils. Finally, cellulose microfibrils fail, leading to complete degradation of PFs. The schematic diagram of the degradation process is shown in Figure 5. Alkali hydrolysis will lead to the decomposition of the hemicellulose and amorphous regions of the cellulose fiber chain, and the integrity of the fiber–matrix interface will be lost, which will affect the performance of the PF-reinforced cement composite.

Correia et al. [72] also found that alkaline degradation of hemicellulose significantly reduced the degree of ground polymerization. The geopolymer matrix showed good adhesion to cellulose fibers without significant degradation. The results of this study will contribute to a better understanding of the role of cellulose in PF-reinforced geopolymers and should serve as a basis for engineering applications. Figure 6 further illustrates the degradation process of PFs. The figure shows the microstructure morphologies of the PFs in the matrix at various stages of aging during the wetting and drying cycles of 0–50 cycles. Figure 6a shows the degradation of lignin and partial hemicellulose; Figure 6b shows the degradation of hemicellulose; Figure 6c shows the stripping of the cellulose phase microfibril; Figure 6d shows the alkaline hydrolysis of the non-crystalline region of the cellulose chain. Since cellulose is the main structural component of PFs, there is no obvious degradation of the mechanical properties in the first three processes.

**Figure 6 molecules-28-01868-f006:**
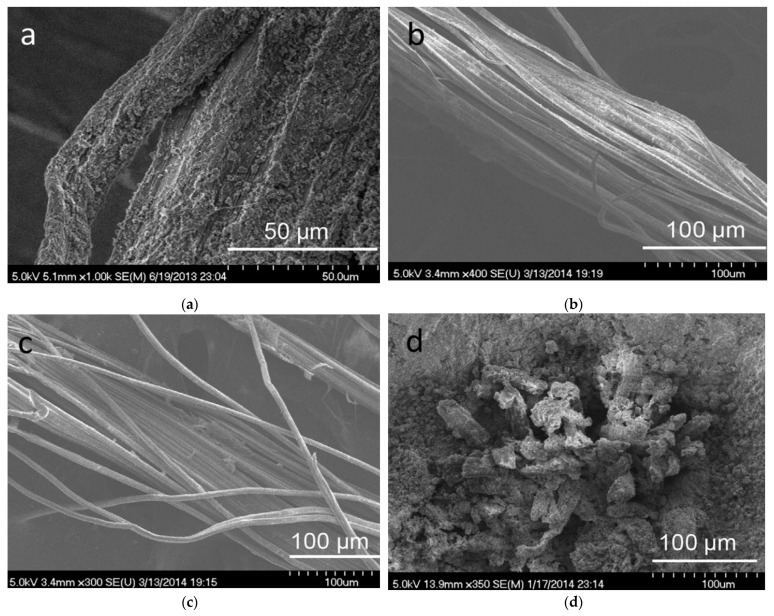
Degradation microstructure of sisal fiber in matrix. (**a**) Degradation of surface lignin, (**b**) Deterioration of hemicellulose, (**c**) Stripping of cellulose, (**d**) Alkaline hydrolysis of cellulose [73].

On the one hand, plant fiber (PF) has good compatibility with the geopolymer matrix, but on the other hand, the fiber in the matrix will degrade. Compared to the cement matrix, the degree of PF degradation in the geopolymer matrix is relatively weak. However, the mechanical properties of the PF reinforcement in the geopolymer matrix are also affected by the degradation of the alkaline matrix. Ye et al. [74] found that cellulose, hemicellulose, and lignin with a mass content of 5% could effectively improve the bending strength of geopolymers. In fact, the cellulose in the PF makes the matrix structure more dense and has fewer pores. The toughness of the composite material increases, and the matrix and fiber also show a good bond without significant degradation. Hemicellulose and lignin cause the porosity of the composites, and their alkaline degradation reduces the degree of polymerization of the composites [27,28].

Table 2 summarizes the chemical composition of PFs. It can be seen that the composition of different PFs is very different. This difference directly affects the alkaline degradation behavior of fibers in the matrix. Generally, the cellulose content of straw and other recycled fibers is lower, and the degradation rate is faster than that of cotton and flax fibers.

**Table 2 molecules-28-01868-t002:** Chemical composition of PFs.

PFs	Cellulose	Hemicellulose	Lignin	Others	Ref.
Cotton	89.7	1.0	2.7	6.6	[74]
Pulp	82.7	2.6	9.8	4.9	[72]
Jute	69.3	14.1	13.6	3.0	[73]
Flax	69.1	17.2	5.4	8.3	[37]
Sisal	68.2	13.7	12.3	5.8	[45,57]
Wheat straw	38.0	36.0	22.0	4.0	[74]
Bamboo	31.5	15	15	38.5	[21,54]
Coir	31.0	19.2	29.7	20.1	[54,75]

### 4.2. Degradation of PFs in Alkaline Solutions

The cellular structure of PF is the fundamental reason for its degradation in a high-alkaline cement matrix. It has been shown that the lignin and hemicellulose in PFs are dissolved in an alkaline solution for a long time, leading to fiber fracture and weak strength. Additionally, the entry of strong alkali substances into the matrix into the PF cavity leads to mineralization of the fiber tissue, which reduces the mechanical properties of the fiber. At the same time, the volume of PF expands after absorbing water in the matrix, affecting the overall stability of the PF structure [76,77]. Saulo et al. [78] found through a test that PF increased the strength and stiffness of the composite in 7 days. However, after 28 days, the mechanical properties of the compounds began to decline, also indicating that the degradation of PFs would occur in the alkaline matrix.

On the basis of the alkaline degradation of PF, scientists have carried out special experiments by means of lye immersion. Holt et al. [79] found that the lignin, cellulose, and hemicellulose content of the PFs was reduced after they were immersed in water, saturated lime water, and sodium hydroxide solution. Yan et al. [80] immersed the flax-reinforced composite material in water, seawater, and a solution of 5% sodium hydroxide. After one year of aging, the property test found that the tensile and bending properties of the composite material declined most significantly in 5% sodium hydroxide alkaline solution. John et al. [81] selected coconut husk and sisal fiber for the experimental study and found that after 28 days of immersion in a saturated solution of calcium hydroxide, the tensile strength of both fibers decreased by approximately 50%. Similarly, Filho et al. [24] also conducted relevant studies by exposing coconut husk and sisal fibers to alkaline media. After 420 days of immersion in sodium hydroxide solution, sisal fiber and coir fiber could maintain 72.7% and 60.9% of the original strength, respectively. In the solution of calcium hydroxide, the sisal and coir fibers maintained only 33.7% and 58.7% strength, respectively, after 210 days.

The degradation of the fibers soaked in an alkaline solution is very obvious, and the same is true for fibers embedded in the matrix. Filho et al. [30] found that due to the degradation of fibers in the alkaline cement matrix, the long-term performance of PFs against the alkaline solution from the cementing matrix was poor, resulting in a gradual reduction in the fracture strength and toughness of the composite materials. Wei et al. [82] studied the effect of metakaolin and montmorillonite replacing cement on PF degradation. The degradation process of PFs was directly studied by means of tensile properties, crystallinity, chemical composition, and microstructure.

Many studies have shown that the degradation of PFs can be alleviated by reducing the alkalinity of the matrix pore solution. The correlation between cement hydration and fiber degradation suggests that PFs degradation is effectively slowed by reducing the alkalinity of the pore solution [83]. Correia et al. [72] studied the effects of calcium aluminate cement (CAC), geopolymer, and OPC on pulp fibers. The results showed that the soak in different pulps changed the surface structure of the fibers, removed lignin, and degraded hemicellulose and cellulose to some extent. After being soaked in the geopolymer and OPC slurry, the tensile strength of the fibers decreased by 34% and 70%, respectively. Because of the inherent properties of PFs, although geopolymers do not contain calcium hydroxide, the high alkalinity of their activators also causes fiber degradation, but the degree of degradation is much lower than that of cement paste.

FTIR analysis of pulp fibers immersed in different mixtures showed that the chemical structure of the fibers changed after 28 days of immersion in alkaline solution [72]. Figure 7a shows the pulp spectra of OPC, CAC, and the geopolymer. The spectra show the characteristic peaks of pulp cellulose fibers. According to further analysis by the researchers, the pulp control curves showed intermolecular hydrogen bonds at 3278 cm^−1^ and 3269 cm^−1^, respectively, and intramolecular hydrogen bonds at 3399 cm^−1^ and 3394 cm^−1^, respectively. See Figure 7b. Compared with the control curve, the immersion of fibers in the alkaline substrate environment (OPC, CAC, and the geopolymer) showed a slight shift toward a lower wave number. Studies have shown that calcium hydroxide in cement migrates to the lumen and wall of the fibers. Although CAC and geopolymers do not contain calcium hydroxide, the high alkalinity of the matrix also changes the hydrogen bonds in the fibers and accelerates the degradation process of lignocellulosic fibers. Changes in the chemical composition and molecular structure of fibers also directly affect the strength of the pulp.

### 4.3. Degradation of PFs in Wetting and Drying Cycles

In engineering practice, composites are in different aging environments during service. These environments have an important effect on the degradation of PFs in the composite matrix. Under different temperature and humidity conditions, the drying and wetting cycles have a great influence on the properties of composites, and there are many related studies.

As mentioned above, different matrix materials of composite materials have a great influence on fiber degradation. Resistance to the wetting and drying cycles is an important index of composite materials. Melo Filho et al. [29] observed that the fiber degradation process occurred rapidly in Portland cement (PC) composites. After 10 cycles of wetting and drying, the bending behavior of the composites changed significantly. After 25 cycles, the residual bending parameters of the composites were basically the same as those of the unreinforced matrix. To analyze the degradation of fibers in the matrix, Mohr et al. [84] found that after 25 wetting and drying cycles, the binary composite containing 90% slag, 30% metakaolin, or 30% silica ash in cement showed no signs of degradation. The cement containing 70% slag + 10% metakaolin or 70% slag + 10% wollastonite ternary blends also effectively prevented degradation. Wei et al. [85] found through tests that due to sisal fiber degradation, the bending strength and toughness of fiber-reinforced cement mortar decreased by 90% and 98%, respectively, after 10 wetting and drying cycles. Using natural diatomaceous earth instead of 20% cement consumed 24.4% calcium hydroxide. After 10 wetting and drying cycles, the bending strength and toughness of the control group were 5.3 times and 7.9 times lower, respectively. After 20 wetting and drying cycles, the tensile strength of the fibers increased 3.8 times.

Trindade et al. [86] concluded that after 15 wetting and drying cycles of jute-reinforced geopolymers, the first crack strength of the composites decreased, but the ultimate strength did not change significantly. A variety of cracks were formed after accelerated cyclic aging of wetting and drying. After 15 cycles, there was no obvious degradation of the fibers, indicating that the reinforced jute geopolymer had superior durability compared to cement-based composites. Rachel et al. [87] found that the ductility of the bagasse-fiber-reinforced geopolymer composite improved after 20 wetting and drying cycles. The wetting and drying cycle is shown to improve matrix bonding and thus improve the ductility of the composites. Santos et al. [45] found that after 10 periods of accelerated aging during wetting and drying, the bending strength of sisal-fiber-reinforced geopolymers reached 11 MPa, while the bending test strength of unaged composites was 15 MPa, and the strength decreased slightly. At the same time, it was found that sisal fiber almost did not degrade after 3 years of natural aging.

The composition and properties of the matrix have a great influence on the degradation of the fiber. Wei et al. [57] showed that compared with cement mortar, the initial first crack strength of the geopolymer composite with metakaolin (MK) was improved. The strength of the 10% MK and 30% MK substitute cement composites was 9% and 34% higher than that of cement mortar, respectively. The tensile strength of the fibers embedded in PC and 10% MK was lower than that of the control group. The decrease in the strength and the change in the slope of the elastic region indicate that the natural fiber has higher stiffness and brittleness in alkaline hydrolysis. Hemicellulose and lignin in natural fibers increase the space between fibers, disperse fibers, and, on the other hand, protect cellulose chains from direct corrosion from high-alkalinity pore solutions. It decreased with increasing MK replacement with PC, indicating that MK effectively reduced sisal fiber degradation and improved the durability of composite materials by reducing the alkalinity of the pore solution.

Table 3 shows the composition and main parameters of PF-reinforced composites under wetting and drying cycles. It can be seen that whether it is natural minerals, such as kaolin, or industrial wastes, such as slag, replacing cement with such silicoaluminate materials can reduce the alkalinity of the matrix and effectively slow down the degradation of fibers. In addition, the effect of the mix proportion of geopolymers on fiber degradation lies in the effect of alkaline activator on fiber degradation. In engineering practice, the amount of alkaline activator should be reduced as much as possible, or other substitutes should be used.

Wei et al. [82] analyzed the load–deflection curves of sisal fiber in cement and geopolymer slurry. Figure 8 shows the load–deflection behavior of the sample under wetting and drying cycles. The maximum peak strength of fiber-reinforced geopolymer is 49.7% higher than that of PC, and the toughness of the geopolymer is 77.71% higher than that of cement. After 5, 15, 30, and 50 cycles, the strength of fiber in the geopolymer decreases by 24.72%, 68.34%, 86.52%, and 91.91%, respectively. After 50 cycles, the cracking strength of the fiber-reinforced geopolymer is 4.17 MPa, 67.34% lower than its initial strength.

### 4.4. Alkaline Degradation of Fiber at High Temperature

In addition to moisture, the mechanical properties of fibrous materials are also extremely sensitive to temperature. As mentioned above, temperature can affect the movement of molecules, and the movement of molecular chains affects the macroscopic mechanical properties of the materials, showing different mechanical properties at different temperatures. Therefore, it is necessary to study the degradation performance of fiber materials in a humid and hot environment, which is a complicated environment. It is well known that natural fibers will encounter more severe attacks in alkaline solutions at higher temperatures, which promotes the hydrolysis of non-formable materials in the fibers, thus limiting their thermal stability [40]. The processing temperature of most natural fibers is limited to around 200 °C [59].

Compared with cement-based materials, the durability of geopolymer materials is reflected in their better resistance to high temperatures, which can protect fibers and slow down degradation. A study by Alomayri et al. [88] showed that fly-ash-based polymer could effectively prevent the degradation of cotton fabric at high temperature. Alomayri’s team also tested geopolymers containing 0.83% cotton fabric mass at 200 °C, 400 °C, 600 °C, 800 °C, and 1000 °C. The flexural strength and fracture toughness of the geopolymer decreased with increasing temperature. The results show that high temperature degrades the cotton fibers and makes the composites brittle. When the temperature is above 600 °C, the mechanical properties of the composites decrease significantly, indicating that fiber degradation is serious. In addition, it was found that after cotton fiber was added, no cracks were found on the surface of the geopolymer at the same temperature. It is shown that the small cavity caused by the degradation of cotton fiber is very effective in preventing the cracking of the matrix due to high temperature. The pores and small channels generated by the degradation of cotton fibers can reduce the pressure of the steam inside the matrix, thus reducing the possibility of cracking [89].

Assaedi et al. [51] showed that the flax fibers degrade significantly at 300 °C. The first transformation of flax fiber degradation occurs at about 25 to 240 °C, when the free water within the fiber evaporates. The second transition occurs at 240–365 °C, when the highest weight loss occurs due to the decomposition of cellulose. Alzeer et al. [90] also verified the loss of mass of the flax fiber at 240–340 °C. The final transition occurs at 365 °C, when the flax fibers begin to break down. At this temperature, flax fibers are free from volatile substances, so there is little mass loss. As can be seen in the figure, due to the evaporation of physically absorbed water, the weight loss rate of the fiber-reinforced geopolymer composite at 260 °C is 10.5%. When the temperature reaches 300 °C, the total weight of the compound decreases further to 15%, indicating that a large amount of fiber degradation occurs within the compound at this temperature.

In fact, PF types differ in their resistance to environmental degradation. Matuana et al. [91] found that due to its hydrophobicity, lignin was relatively resistant to microbial degradation and acid degradation but sensitive to ultraviolet degradation. The hollow structure of PF promotes the absorption and penetration of more water into the surrounding fiber cells, promoting fiber degradation. Due to the lower lignin content in flax fibers, flax has a relatively higher thermal stability than jute and sisal fibers [92]. Methacanon et al. [61] studied the performance of reinforced geotextiles produced with reed, sisal, water hyacinth, and coiled hemp fibers and found that reed fibers and water hyacinth fibers contained a higher content of hemicellulose and could absorb more water, which could lead to thermal degradation at lower temperatures. In their test sample, Bui et al. [67] observed no decomposition peak after five drying and wetting cycles in the temperature range of 270 °C to 300 °C, indicating that the coir fibers in the gel matrix did not undergo thermal degradation and were naturally degraded.

### 4.5. Mineralization of PFs in the Matrix

Generally speaking, fiber degradation of PF-reinforced cement-based composites is accompanied by fiber mineralization. During the matrix hydration process, the Ca, Mg, Al, and silicon plasma impregnated fiber cell walls were defined as fiber mineralization [58]. As a result of the migration of hydration products such as calcium hydroxide into the PF cavity, the fiber wall, and interfiber void, the toughness of the composite decreases. Filho et al. [24] evaluated sisal and coconut exposure to alkaline calcium and sodium hydroxide solutions and found that sisal and coconut fibers stored in calcium hydroxide solutions at pH 12 completely lost strength after 300 days. This is due to the crystallization of the Ca(OH)_2_ solution in the fibrous structural cavity, leading to alkaline degradation and mineralization of the fibers. Wei et al. [57] found that the absorbency of fibers in Ca(OH)_2_ and sodium hydroxide solutions was completely different due to the influence of mineral penetration and precipitation rich in Ca ions. Due to migration of the Ca(OH)_2_ solution in cement, crystallization is caused in lumen cells and PF voids, gradually losing the strength and toughness of the fibers, resulting in a substantial reduction in their strength ability [81]. Filho et al. [30] also demonstrated that PFs are mineralized by the hydration product C-S-H gel. In addition to alkaline hydrolysis of PF, fiber mineralization is also an important cause of loss of fiber strength and strain capacity [93]. With the increase in fiber water absorption, the penetration of the product into the fiber cavity and the intermediate lamellar fraction increased, accelerating the fiber mineralization and embrittlement behavior. Wei et al. [45] reported that dynamic wetting and drying cycles have more accelerated effects on alkali hydrolysis and cell wall mineralization of the amorphous components of PF fiber, with the highest crystallinity index and the lowest cellulose content.

In addition to the increased mineralization of sisal and coconut fiber reinforcement, other reinforced fiber mineralization studies are also being carried out. Olayiwola et al. [94] found that there was particle mineralization and partial degradation of hemicellulose in the alkaline matrix of geopolymers in black robinia pseudoacacia and longleaf acacia, as well as bagasse. In summary, in cement composites, the mineralization of PFs was caused by precipitation of fiber cells and surface calcium hydroxide, and the degradation of cellulose, hemicellulose, and lignin was caused by adsorption of calcium and hydroxyl ions. For geopolymer composite fibers, no mineralization was observed due to the low calcium hydroxide content in the matrix [31].

## 5. The Path to Slow Fiber Degradation

Currently, in order to alleviate the degradation characteristics of PF in the alkaline matrix, researchers have taken a variety of measures to carry out relevant research. The alkaline matrix is usually modified to consume the content of the calcium hydroxide alkaline component produced during cement hydration [95,96,97,98,99,100,101]. The other method is to modify the fiber to improve the property of the interfacial bond and the compatibility between the fiber and the matrix [102,103,104,105,106,107,108,109,110,111,112,113,114,115,116,117,118,119]. In addition, increasing the toughness of the composites and adding nanomaterials can also improve the reinforcing properties of the fibers in the matrix [120,121,122,123,124].

### 5.1. Improve the Alkaline Environment of the Matrix 

As summarized in this paper, substituting geopolymer for cement is an effective way to slow the degradation of the PF reinforcement in the matrix and prevent fiber mineralization. The amount of alkaline activator of the geopolymer was further reduced, and the substitution of the alkaline activator was adopted. In addition, the method of adding mineral admixture and accelerating carbonization can also be used.

#### 5.1.1. Accelerated Carbonization

For PF-reinforced cement-based composites, the purpose of carbonization is to make the calcium hydroxide hydration product of the cement react with carbon dioxide to produce calcium carbonate. Tonoli et al. [95] conducted accelerated carbonization tests using sisal and cowhide pulp-fiber-reinforced cement-based materials. After carbonization, the toughness of the composite material increased by 80%, and the degradation of the fiber in the matrix was reduced. Many studies have shown that carbonization can reduce the porosity and water absorption of composite materials and improve the degree of binding between the fiber and the cement matrix interface [76,96]. Due to the chemical stability of the carbonized products and the reduced capillary porosity, the adhesion between the matrix and the fiber is improved, and degradation of the fiber is delayed. 

#### 5.1.2. Add Mineral Admixtures to the Matrix

In engineering practice, mineral admixtures are often added to reduce the content of calcium hydroxide in the matrix. When mineral admixtures, such as silica fume, blast furnace slag, and fly ash, are added to cement materials, secondary hydration reactions can occur with calcium hydroxide in cement to produce calcium silicate hydrate or calcium aluminate hydrate [97]. When the calcium hydroxide content in the matrix is reduced, deterioration of the PF properties in the matrix is avoided, and the strength and toughness of the composite materials are guaranteed. The incorporation of clay minerals can also reduce the alkalinity of the pore solution, thus significantly alleviating alkali hydrolysis and mineralization of PF [98,99]. Multiple studies have agreed that the use of auxiliary mineral materials as cement substitutes to reduce substrate alkalinity can prevent chemical attacks by lignocellulosic fibers on the substrate [100,101].

### 5.2. Modification of PFs

In order to improve the properties of PFs in the matrix, modification is usually used. Treatment with fiber modification can better improve fiber properties and effectively improve fiber resistance to the risk of degradation [102,103]. Pretreatment of natural fiber is an effective method to improve fiber degradation resistance. The treatment methods include silane coating [104,105], keratinization [106], sodium silicate or potassium silicate [107], bacterial nanocellulose-coated fibers [108], alkaline treatment, etc. The mechanical properties of the fibers are improved by forming a protective layer on the fiber surface or strengthening the cellulose structure of the fibers [109]. Fiber modification mainly includes physical modification, chemical modification, and biological modification, among which chemical modification is the most commonly used [63,110].

Chemical modification of fibers is carried out to remove easily degradable hemicellulose, lignin, pectin, and other substances on the surface, so that they have a relatively rough apparent morphology, and the interface between the fibers and the matrix forms a mechanical interlock morphology [111,112]. Fiber treated with chemical modification has low water absorption. Table 4 shows the summary of the treatment techniques of PFs.

As can be seen from Table 4, different treatment methods have different effects on fiber properties. Tonoli et al. [113] modified the eucalyptus fiber with silane, which reduced water retention and showed good dimensional stability. Ardanuy et al. [114] treated the three fibers of canna hemp, agave, and sisal by adding 5% styrene-acrylic copolymer, and the water absorption rate of the composite was reduced by 50%, and its stiffness and dimensional stability were improved. After modification, a dense and condensed transition zone is formed at the fiber–matrix interface to make the fiber adhere to the interface and prevent fiber mineralization [115]. Fonseca et al. [27] treated palm, shaving grass, and jute fibers with hot water, keratinization, 8% sodium hydroxide solution, and hybridization, and the treated fibers did not degrade further due to the increase in temperature and pressure in the autoclave. The treatment of the tucum fiber resulted in more connections between the microfibers of the cellulose chain, resulting in the highest crystallinity index (66.73%) and a tensile strength of 318.81 MPa. 

Many cases have been studied on the modification of bast fibers [116]. Kumar et al. [43] modified ramie with NaOH solution to remove the lignin component of the fibers. Maichin et al. [117] treated hemp fibers with different concentrations of sodium hydroxide. At the same time, it was found that the basicity of the geopolymer matrix affected the fiber self-treatment process. After fiber surface modification, the compatibility and cohesiveness between the fiber and the cement slurry are improved. Roy [118] found that the abaca fiber had the highest tensile strength when soaked in a solution of Al_2_ (SO_4_)_3_ at pH 6 for 12 h. The fiber deposition of aluminum compounds makes the surface rougher, improves the interface between the fiber and the matrix, and protects the fiber from thermal degradation. Asante et al. [119] found that, after washing with hot water, the specific strength of the ground polymers of the pine and eucalyptus particles increased by 27% and 3%, respectively. The results showed that the special extract of pine fiber was washed off with hot water, the compatibility between the geopolymer and wood fiber was better, and the interface bonding ability between the fiber and the matrix was significantly improved.

### 5.3. The Influence of Nanomaterials

The addition of nanomaterials can not only accelerate the polymerization process, but the unreacted nanoparticles are also coated by the matrix material and play the role of a filling, making the matrix structure more dense and improving the performance of the composite material [120,121]. 

Usually, the properties of PFs deteriorate as a result of alkali ion attack, and mineralization of PF cell walls leads to brittleness. Nano-SiO_2_ can consume the alkaline solution in the matrix, so the alkali of the composite decreases, thus reducing the degradation rate of PF. Assaedi et al. [122] found that flax-fiber-reinforced geopolymer composites containing nano-SiO_2_ had higher long-term loading capacity. Compared to the fourth week, the bending strength of the nano-SiO_2_ composites was reduced by approximately 10.3% at thirty-two weeks, while the bending strength of the composites without nano-SiO_2_ was reduced by 22.4%. This shows that the degree of fiber degradation of the nanocomposite is lower. Rahman et al. [123] showed that spherical silica nanoparticles prepared with rice husk ash reduced the porosity of the geopolymer by 20%, and the bending strength increased by 27% and 97%, respectively, when rice husk ash and SiC whisker were added. The SiC whisker could effectively improve the matrix bridging ability. Assaedi et al. [124] added 1.0–3.0% nano-clay to enhance adhesion between the matrix and the flax fibers. Hakamy et al. [99] also adopted nanocalcined clay to strengthen the bond between the matrix and the hemp fiber. 

In fact, nanomaterials can accelerate the polymerization reaction of the matrix, increase the number of matrix gels, improve the density of the matrix, improve the bond between the fiber and the matrix, and further reduce the deterioration rate of the PF in cementitious materials.

### 5.4. Influence of Toughening Matrix

In engineering practice, the type and content of the fiber and the surface shape of the fiber all affect the toughness of composite materials. The toughness of the matrix is simply a reflection of the compatibility between the fiber and the matrix, the interfacial bond and the mechanical properties of the PF itself [125]. Assaedi et al. [51] found that the addition of flax fiber significantly improved the flexural strength and fracture toughness of the composites and reduced the degradation of flax fiber. Mazen et al. [126] found that the bending strength of the luffa fiber-reinforced geopolymer composite increased from 3.4 MPa to 14.2 MPa compared to the unreinforced material. After 20 months of aging, the bending strength of the 10% luffa fiber composite increased from 8.6 to 9.8 MPa. Due to the toughness of the material, the long-term performance of the composite did not decrease significantly. Mourak et al. [127] reduced the porosity of the composite jujube stem fiber and increased the mechanical strength and density. The bending strength of the composite increased from 1.7 to 17.4 MPa, and the water absorption increased from 11.3 to 19%. The toughening effect is explained further. Chen et al. [128] found that the main function of the sweet sorghum fiber was to improve the bending property of the matrix and control the further development of cracks in the matrix, thus accelerating the degradation of the sweet sorghum fiber. At the same time, fibers can also form an impervious layer in the matrix, making the matrix structure more dense [129].

The good adhesion between the PF and the matrix increases the toughness of the compound. In this way, the development of microcracks can be inhibited and stabilized in the alkaline matrix, and the contact between the fiber and harmful substances in the external environment can be prevented, thus slowing down the deterioration of the fiber performance.

## 6. Conclusions

In this paper, the alkaline degradation mechanism, the factors influencing the degradation process, and the improvement path of PF in the alkaline matrix were summarized. The advantages of PF are obvious: wide source, cheap price, energy savings, and environmental protection. However, the structure and properties of PFs lead to degradation of the alkaline matrix. Improving the durability of PF-reinforced geopolymers and slowing down the degradation in the matrix are the key factors for their large-scale engineering applications. The main conclusions are as follows.

The hygroscopicity of PF affects its mechanical properties. However, this property can be used to achieve the curing performance in the matrix of composite materials, to improve the strength of the matrix. 

There is a certain degree of alkaline degradation of PF in the geopolymer matrix, and the degree of degradation is closely related to the alkaline strength of the geopolymer. Due to the influence of calcium hydroxide, PF results in both degradation and mineralization in the cement matrix. Compared to the cement matrix, the geopolymer matrix significantly slowed the degradation of the PFs. 

The most direct way to reduce the degradation of PFs in the matrix is to reduce still the pH value and calcium hydroxide content of the matrix. 

Alkaline degradation of PFs in the matrix has an adverse effect on the mechanical properties of composite materials, and the effect of fiber degradation can be mitigated by chemical modification.

Nanomaterials can improve the microstructure of the composite matrix, accelerate the polymerization reaction of the matrix, increase the amount of matrix gel, improve the matrix density, improve the bond between the fiber and the matrix, and thus reduce the rate of deterioration of PF. 

In fact, the PFs used in the matrix are classified as recycled and virgin. Straw fibers and coir fibers are wastes. It is of environmental significance to make full use of waste fiber. In addition, substituting geopolymer for cement is an effective way to slow the degradation of PFs in the matrix. At the same time, a low basic activator substitute should be used in the geopolymer matrix. In previous research works, there has been relatively little quantitative study on the mix proportion of the geopolymer influencing the degradation of the fiber, so this should be one of the research directions in the future.

## Figures and Tables

**Figure 1 molecules-28-01868-f001:**
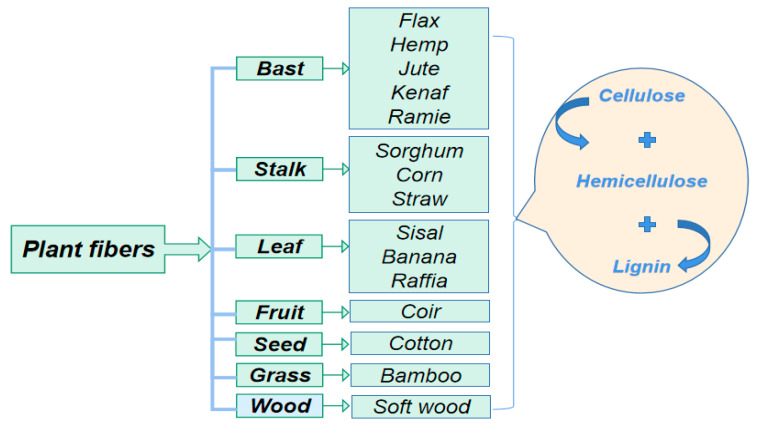
Classification of commonly used PF reinforcements.

**Figure 2 molecules-28-01868-f002:**
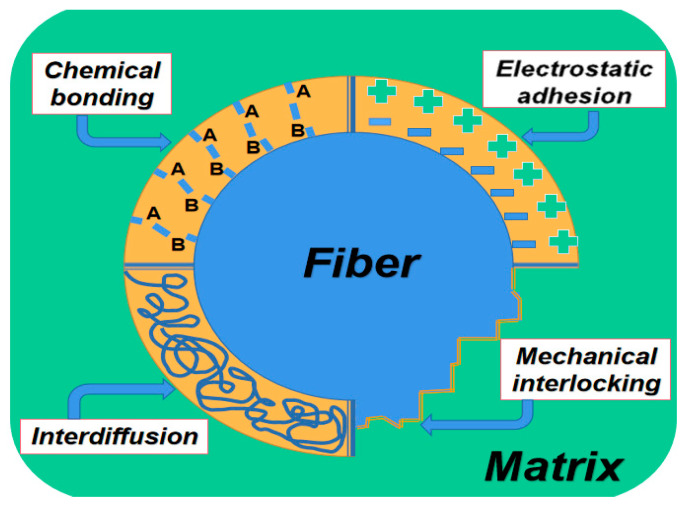
Form of interfacial adhesion between fiber and matrix.

**Figure 3 molecules-28-01868-f003:**
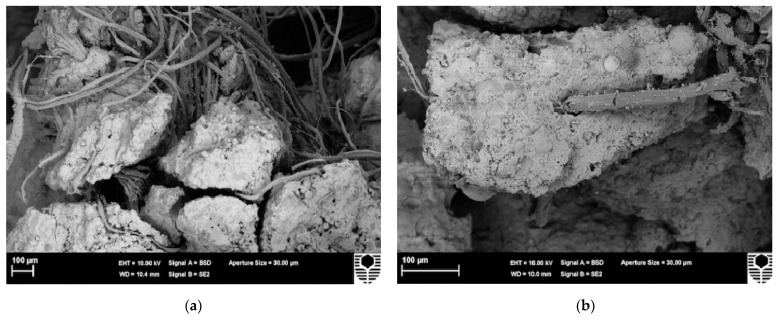
Microscopic morphology of interfacial adhesion between fibers and matrix. (**a**) Fiber bridging crack and fiber debonding; (**b**) Fiber adhesion to matrix and fiber fracture [51].

**Figure 4 molecules-28-01868-f004:**
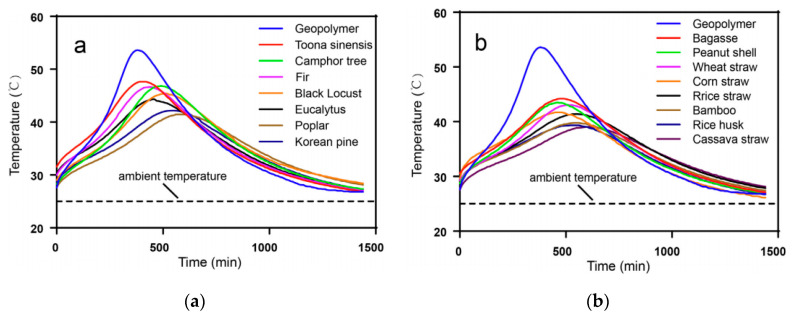
Curve of polymerization temperature of PF geopolymer with time. (**a**) Wood fiber; (**b**) Non-wood fiber [65].

**Figure 5 molecules-28-01868-f005:**
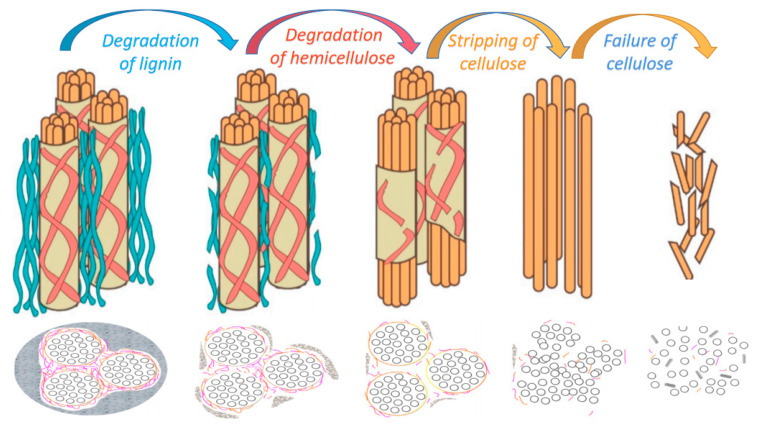
The alkaline degradation mechanism of PFs [37,57].

**Figure 7 molecules-28-01868-f007:**
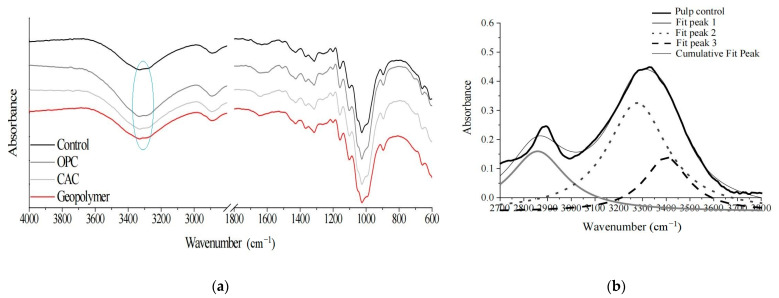
Pulp FTIR analysis of OPC, CAC, and geopolymer. (**a**) FTIR spectra of pulp in the control condition and after 28 days of immersion in the OPC, CAC, and geopolymer pastes; (**b**) Deconvoluted FTIR spectra of pulp control condition [72].

**Figure 8 molecules-28-01868-f008:**
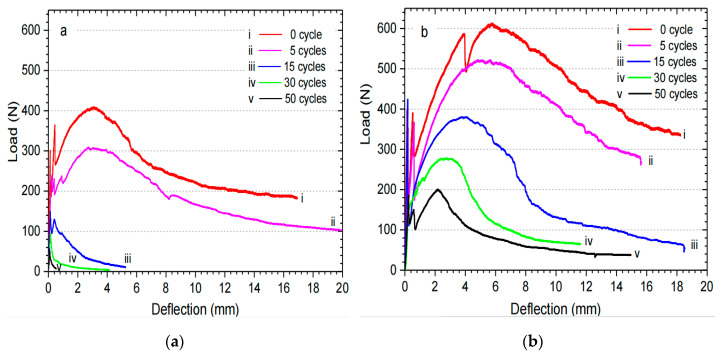
Typical load–deflection curves under wetting and drying cycles. (**a**) Cement slurry; (**b**) Geopolymer [82].

**Table 3 molecules-28-01868-t003:** Research on PF reinforcements under wetting and drying cycles and main parameters.

PFs	PF Content	Binder	Cure Condition	Evaluation Method	Number of Cycles	Evaluation Time	Degradation Mechanism	Ref.
Sisal	2% volume fraction	PC-10% MK; 30% MK	Immersed in CH-saturated water at 23 ± 2 °C	Flexural; Separation approach	5; 15; 30	7 days; 28 days	Degradation and mineralization	[57]
Sisal	6% volume fraction	PC-50% MK	100% RH, 23 ± 1°C	Flexural; Fracture behavior	5; 10; 15; 20; 25	28 days; 180 days; 1 year; 5 years	Degradation and mineralization	[31]
Kraft pulp	4% volume fraction	Binary composite; Wollastonite ternary blend	Immersed in limewater	Flexural strength; Post-cracking toughness	25	28 days	Degradation	[84]
Sisal	1% volume fraction	PC-5% DE; 10% DE; 15% DE; 20% DE	Immersed in saturated lime water at 23. 2 °C	Tensile Strength; TGA analysis	5; 10; 15; 20	28 days	Degradation and mineralization	[85]
Jute	10% of volume fraction	SF-, MK-, and BFS-based geopolymer	At room temperature 25 ± 2°C	Tensile and flexural tests	15	7 and 28 days	No obvious degradation	[86]
Sugarcane bagasse	1.5%, 3%, 4.5%, 6%, and 7.5% mass fraction	Laterite-based geopolymer	At room temperature	Mass loss; Compressive strength loss	5; 10; 20	28 days	Degradation	[87]
Sisal	2% mass fraction	Sludge-based geopolymer	At room temperature 27 °C and 80% RH	Flexural strength	10	6 months or 3 years	No degradation	[45]

Portland cement—PC; Metakaolin—MK; Diatomaceous earth—DE; Relative humidity—RH; Silica fume—SF; Blast furnace slag—BFS.

**Table 4 molecules-28-01868-t004:** Summary of the treatment techniques of PFs.

PF Types	Treatment Methods	Modification Types	Measurements	Authors
Palm; Shaving grass; Jute	Hot water; Keratinization; 8% NaOH solution; Hybridization	Physical; Chemical	No further degradation; Crystallinity index; Tensile strength	Fonseca et al. [27]
Eucalyptus	Silane	Chemical	Water retention; Dimensional stability	Tonoli et al. [113]
Canna hemp; Agave; Sisal	5% styrene-acrylic copolymer	Chemical	Water absorption; Stiffness; Dimensional stability	Ardanuy et al. [114]
Ramie	NaOH solution	Chemical	Lignin removal	Kumar et al. [43]
Hemp	NaOH of different concentrations	Chemical	Compatibility; Cohesiveness	Maichin et al. [117]
Abaca	Al_2_ (SO_4_)_3_ solution at pH 6	Chemical	Tensile strength; Rougher surface	Roy [118]
Pine; eucalyptus	Hot water	Physical	Specific strength; Compatibility	Asante et al. [119]

## Data Availability

Not applicable.

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
