# Peer review of "Alkaline Degradation of Plant Fiber Reinforcements in Geopolymer: A Review"

_molecules, 2023, doi:10.3390/molecules28041868_

Round 1

Reviewer 1 Report

Reviewer comment: The manuscript presents a review of the alkaline degradation of plant fiber reinforcements in geopolymer. Plant fibres (PFs) are numerous, cheap, lightweight, biodegradable, matrix-adhesive, and have several applications as reinforcements.. Overall, the manuscript is well organized and well structured. However, the paper should be improved according to the comments in the attached file.

Author Response

Reviewer 1:

 The manuscript presents a review of the alkaline degradation of plant fiber reinforcements in geopolymer. Plant fibres (PFs) are numerous, cheap, lightweight, biodegradable, matrix-adhesive, and have several applications as reinforcements.. Overall, the manuscript is well organized and well structured. However, the paper should be improved according to the comments in the attached file.

Thank you very much for reviewing our manuscript in your busy schedule. Based on your valuable suggestions, we have carefully revised our manuscript.

1-  Please cite more previous studies.

Thank you very much for your review. According to your suggestion, we have cited more previous studies. (The revised page 2, line 63; The revised page 19, lines 735-738).    

2- Please discuss the role of these fiber on compressive strength.

Thank you very much for your review. According to your suggestion, we have carefully discussed the role of these fiber on compressive strength. (The revised page 2, lines 85-88)

3-  reference is required here. 

Thank you very much for your review. According to your suggestion, we have cited the following reference: https://doi.org/10.3390/polym14245504 and https://doi.org/10.1016/j.ceramint.2022.03.103.  (The revised page 2, line 95; The revised page 19, lines 745-748)

4-  Please provided the references.

Thank you very much for your review. According to your suggestion, we have provided the references. (The revised page 3, line 107; The revised page 20, lines 749-750)

5- Pleasecite reference and provide similar recent studies.

Thank you very much for your review. According to your suggestion, we have carefully  cited reference and provided similar recent studies. (The revised page 9, line 234; The revised page 20, lines 763-765)

6- Please mention the peak values of IR.

Thank you very much for your review. According to your suggestion, we have carefully revised the content. (The revised page 10, lines 351-353)

7- Please cite reference and provide similar recent studies.

Thank you very much for your review. According to your suggestion, we have carefully revised the content. (The revised page 15, lines 509-516)

In addition, we have also revised other parts of the article according to your review suggestions (highlighted parts in the manuscript).

According to your suggestion, we have carefully and comprehensively revised the manuscript (both in the content and in the language).

Finally, thank you again for your wonderful review of our article in your busy schedule.

Reviewer 2 Report

- How does the mix proportion of geopolymer influence the degradation of the fiber?

- The figures in the manuscript are directly taken from others' work. It is better to compare and summarize researchers' results for the degradation in forms of tables or figures.

- What are the main factors affecting the degradation of fibers in the geopolymer?

- The authors should pay more attention on the possible approaches for impoving the behavior of composite with fibers in geopolymer.

Author Response

Reviewer 2:

- How does the mix proportion of geopolymer influence the degradation of the fiber?

Thank you very much for reviewing our manuscript in your busy schedule. Based on your valuable suggestions, we have carefully revised our manuscript. (The revised page 11, lines 396-402; The revised page 18, lines 654-658; The revised page 18, lines 668-674).

- The figures in the manuscript are directly taken from others' work. It is better to compare and summarize researchers' results for the degradation in forms of tables or figures.

Thank you very much for reviewing our manuscript in your busy schedule. Based on your valuable suggestions, we have carefully revised our manuscript.

Added Table 2 ( the revised page 9, line 291);

Added Table 3 (the revised page 12, line 411);

Added Table 4 (the revised page 16, line 565)

Figure 9 was deleted.

Figure 5 and Figure 7 have been modified. ( the revised page 7, line 249; the revised page 10, lines 351-353)

- What are the main factors affecting the degradation of fibers in the geopolymer?

Thank you very much for your review. According to your suggestion, It is explained in 4.1. Mechanism of degradation of PF in the matrix. Figure 5 has also been supplemented and modified. (The revised page 7, line 248; The revised pages 7-8, lines 252-259).  

- The authors should pay more attention on the possible approaches for impoving the behavior of composite with fibers in geopolymer.

Thank you very much for reviewing our manuscript in your busy schedule. Based on your valuable suggestions, we have carefully revised our manuscript.(The revised page 15, line 508; The revised page 17, lines 651-656; The revised page 18, lines 665-669)

In addition, we have also revised other parts of the manuscript according to your review suggestions (highlighted parts in the manuscript).

The latest references have been added.

We are very sorry for the trouble caused to your review due to our bad writing. According to your suggestion, we have carefully and comprehensively revised the manuscript (both in the content and in the language).

Finally, thank you again for your wonderful review of our article in your busy schedule.

Reviewer 3 Report

The term "A review" must be added to the title

the last paragraph of the introduction requires to be written again to show the main objectives of the research.

It is important to show the gap in previous researchers

Conclusions requires to be enhanced and related to the main ideas of the research. 

Author Response

Reviewer 3:

- The term "A review" must be added to the title

Thank you very much for reviewing our manuscript in your busy schedule. Based on your valuable suggestions, we have carefully revised our manuscript. (The revised page 1, lines 1-2).

- the last paragraph of the introduction requires to be written again to show the main objectives of the research.

Thank you very much for reviewing our manuscript in your busy schedule. Based on your valuable suggestions, we have carefully revised our manuscript.(The revised page 2-3, line 98-106)

- It is important to show the gap in previous researchers.

Thank you very much for reviewing our manuscript in your busy schedule. Based on your valuable suggestions, we have carefully revised our manuscript.(The revised page 2, lines 97-99)

- Conclusions requires to be enhanced and related to the main ideas of the research. 

窗体底端

Thank you very much for reviewing our manuscript in your busy schedule. Based on your valuable suggestions, we have carefully revised our manuscript.(The revised page 17, lines 651-656; The revised page 18, lines 665-669).

In addition, we have also revised other parts of the manuscript according to your review suggestions (highlighted parts in the manuscript).

According to your suggestion, we have carefully and comprehensively revised the manuscript (both in the content and in the language).

Finally, thank you again for your wonderful review of our article in your busy schedule.

Reviewer 4 Report

This is one of the best review papers on the use of plant-based fibers in geopolymer. In my opinion, it can be accepted after minor revision:

1)     In the abstract mention the names of plant fibers which were investigated or analyzed?

2)     It is also encouraged to include a chart showing the cellulusos, hemicellulose, and lignin compoments/percentages for different fibers.

3)     The color combination/background should be white for Fig. 1 and Fig. 2. Choose a format suitable for B&W printing.

4)     There is deficiency on the information on the discussion banana fiber and jute fiber reinforced concretes. Recently published studies are ingnored : 1) https://www.sciencedirect.com/science/article/abs/pii/S2352710222010191; 2) https://www.sciencedirect.com/science/article/abs/pii/S2352710222010348; 3) https://www.tandfonline.com/doi/full/10.1080/15440478.2023.2170947;

5)     Also classify the fibers as recycled and virgin if we consider their use in concrete. For instance, coconut fibers are waste, while jute is solely produced for fibers. Including information on the environmental significance would further enhance the quality of this paper.

6)     Figure 9 shows the load-deflection behavior for fiber reinforced concrete, but its not mentioned in the caption of figure.

7)     Provide summary of the treatment techniques in a separate table for the readers benefit.

Author Response

Reviewer 4:

This is one of the best review papers on the use of plant-based fibers in geopolymer. In my opinion, it can be accepted after minor revision:

1)     In the abstract mention the names of plant fibers which were investigated or analyzed?

Thank you very much for reviewing our manuscript in your busy schedule. Based on your valuable suggestions, we have carefully revised our manuscript. (The revised page 1, line 10).

2)     It is also encouraged to include a chart showing the cellulusos, hemicellulose, and lignin compoments/percentages for different fibers.

Thank you very much for reviewing our manuscript in your busy schedule. Based on your valuable suggestions, we have carefully revised our manuscript.(The revised page 9, line 291)

3)     The color combination/background should be white for Fig. 1 and Fig. 2. Choose a format suitable for B&W printing.

Thank you very much for reviewing our manuscript in your busy schedule. Based on your valuable suggestions, we have carefully revised our manuscript.(The revised page 3, line 114; The revised page 4, line 150)

4)     There is deficiency on the information on the discussion banana fiber and jute fiber reinforced concretes

窗体底端

Thank you very much for reviewing our manuscript in your busy schedule. Based on your valuable suggestions, we have carefully revised our manuscript.(The revised page 19, lines 719-720; The revised page 20, lines 830-831; The revised page 19, lines 758-760; ).

5)     Also classify the fibers as recycled and virgin if we consider their use in concrete. For instance, coconut fibers are waste, while jute is solely produced for fibers. Including information on the environmental significance would further enhance the quality of this paper.

Thank you very much for reviewing our manuscript in your busy schedule. Based on your valuable suggestions, we have carefully revised our manuscript.(The revised page 19, lines 655-671)

6)     Figure 9 shows the load-deflection behavior for fiber reinforced concrete, but its not mentioned in the caption of figure.

Thank you very much for reviewing our manuscript in your busy schedule. Based on your valuable suggestions, we have carefully revised our manuscript.(The revised page 12, lines 415-417)

7)     Provide summary of the treatment techniques in a separate table for the readers benefit.

Thank you very much for reviewing our manuscript in your busy schedule. Based on your valuable suggestions, we have carefully revised our manuscript.(The revised page 16, line 565)

In addition, we have also revised other parts of the manuscript according to your review suggestions (highlighted parts in the manuscript).

According to your suggestion, we have carefully and comprehensively revised the manuscript (both in the content and in the language).

Finally, thank you again for your wonderful review of our article in your busy schedule.

Round 2

Reviewer 1 Report

The authors have revised and addressed all the comments provided. Therefore, I recommend publishing it in its current state.

Reviewer 2 Report

All comments are well replied.